

# Rainwater propagation through snowpack during rain-on-snow events under different snow condition

Roman Juras[1,2*], Sebastian Würzer[2], Jirka Pavlásek[1], Tomáš Vitvar[1,3], and Tobias Jonas[2]

[1] Faculty of Environmental Sciences, Czech University of Life Sciences Prague, Kamýcká 129, 165 21, Prague,
Czech Republic
[2] WSL Institute for Snow and Avalanche Research SLF, Flüelastrasse 11, 7260 Davos Dorf, Switzerland
[3] Faculty of Civil Engineering, Czech Technical University in Prague, Thákurova 7, 166 29 Prague 6, Czech
Republic

*Correspondence to*: Roman Juras (juras@fzp.czu.cz)

**Abstract.** The mechanisms of rainwater propagation and runoff generation during rain-on-snow (ROS) are still
insufficiently known. Understanding the behaviour of liquid water within the natural snowpack is crucial
especially for forecasting of natural hazards such as floods and wet snow avalanches. In this study, rainwater
percolation through snow was investigated by sprinkling deuterium enriched water on snow and applying an
advanced hydrograph separation technique on samples collected from the snowpack runoff. This allowed
quantifying the contribution of rainwater and snowmelt in the water released from the snowpack. Four field
experiments were carried out during the winter 2015 in the vicinity of Davos, Switzerland. For this purpose,
large blocks of natural snow were isolated from the surrounding snowpack to inhibit lateral exchange of water.
These blocks were exposed to artificial rainfall with 41 mm of deuterium enriched water. The sprinkling was run
in four 30 minutes periods separated by three 30 minutes non-sprinkling periods. The runoff from the snow
block was continuously gauged and sampled for the deuterium concentration. At the onset of each experiment
initially present liquid water content was first pushed out by the sprinkling water. Hydrographs showed four
pronounced peaks corresponding to the four sprinkling bursts. The contribution of rainwater to snowpack runoff
consistently increased over the course of the experiment but never exceeded 86 %. An experiment conducted on
a cold snowpack suggested the development of preferential flow paths that allowed rainwater to efficiently
propagate through the snowpack limiting the time for mass exchange processes to take effect. On the contrary,
experiments conducted on ripe isothermal snowpack showed a slower response behaviour and resulted in a total
runoff volume which consisted of less than 50 % of the rain input.

Keywords: *hydrograph separation, stable isotopes, sprinkling experiment, preferential flow, flood forecasting*

## 1 Introduction

Rain-on-snow (ROS) events are a natural phenomenon which has been in the focus of hydrological research in
the past decades, particularly because of their high potential to cause natural hazards. ROS initiated severe floods
in the past in many European countries such as Germany (HND Bayern, 2011; Sui and Koehler, 2001),
Switzerland (Badoux et al., 2013; Rössler et al., 2014), Czech Republic (Čekal et al., 2011) or US (Ferguson,
2000; Kattelmann, 1997; McCabe et al., 2007). Rainwater also affects snowpack stability which can initiate
formation of wet snow avalanches (Ambach and Howorka, 1966; Baggi and Schweizer, 2008; Conway and
Raymond, 1993) or trigger slushflows (Hestnes and Sandersen, 1987; Nyberg, 1989; Onesti, 1987). In addition





to natural hazards, ROS events are also relevant from a geochemical point of view. Rainwater affects transport of ions (Jones et al., 1989) and solutes (Feng et al., 2001; Harrington and Bales, 1998; Lee et al., 2008; Waldner et al., 2004) through snow which affects the pH and chemical compositions of adjacent streams (Casson et al., 2014; Dozier et al., 1989; MacLean et al., 1995).

Predicting snowpack runoff for an upcoming ROS event requires the understanding of water transport processes through snow. Water input from heavy rainfall flows typically faster through a snowpack than meltwater outside of rain periods, which is why ROS situations may entail an augmented flood risk (Singh et al., 1998). Interactions between the liquid and solid phase of water make the water flow modelling in snow more difficult compared to other porous media like soil or sand where the solid phase is supposed to be stable. Existing water
flow models for snow have rarely been specifically tested for ROS scenarios, nevertheless Würzer et al. (2016b) have recently introduced a new approach integrated within the SNOWPACK model (Bartelt and Lehning, 2002; Wever et al., 2015).

Presence of liquid water in snow fastens the metamorphism processes such as snow settling, snowpack warming (Conway and Benedict, 1994) and grain coarsening (Gude and Scherer, 1998; Tusima, 1985). These processes
entail a higher hydraulic conductivity and snow permeability which lead to faster water flow (Calonne et al., 2012; Conway and Benedict, 1994). Rainwater introduced to the snowpack during ROS represents an important additional source of liquid water besides snowmelt which can contribute to the generation of snowpack runoff.

There is still a lack of knowledge how rainwater is propagating though snow to generate snowpack runoff and what runoff portion can rainwater represent under various snow conditions. Previous studies have shown that
water can flow through snow in two different regimes, matrix flow and preferential flow, which are both governed by specific snow properties (Schneebeli, 1995; Waldner et al., 2004). In the matrix flow regime snow is wetted top down uniformly with all snow being wet above the wetting front (Schneebeli, 1995; Techel et al., 2008). Preferential flow, on the other hand, is characterised by spatially heterogeneous wetting patterns with horizontally isolated wet and dry zones often referred to as flow fingers (e.g. Techel et al., 2008; Waldner et al.,
2004). These patterns grow with percolation intensity and grain size (Hirashima et al., 2014). During dye tracer experiments in non-ripe snowpack with temperatures below the freezing point, matrix flow was observed in the uppermost layers of the snowpack whereas preferential flow was observed in deeper layers only (Würzer et al., 2016b, Techel et al., 2008). Concepts of water flow behaviour in snowpack were further investigated in various approaches including rainfall simulation (Conway and Benedict, 1994; Eiriksson et al., 2013; Juras et al., 2013;
Singh et al., 1997), artificial wetting (Avanzi et al., 2015; Katsushima et al., 2013; Yamaguchi et al., 2010) or numerical modelling (Hirashima et al., 2010, 2014, Wever et al., 2014, 2015).

The fact that rain and melting snow feature a different isotopic content can be used to differentiate between both components in the snowpack runoff analogically to hydrograph separation, which is a widely used technique especially in watershed hydrology (Buttle et al., 1995; Dinçer et al., 1970; Unnikrishna et al., 2002). Snowpack
usually features a heterogeneous vertical isotope composition (Lee et al., 2010; Zhou et al., 2008) which is partially homogenized over the course of the winter season by a combination of moisture exchange, meltwater and rain infiltration (Krouse et al., 1977; Unnikrishna et al., 2002). Isotopically lighter meltwater is produced at the beginning of snowmelt and becomes heavier as melt progresses. This change is augmented by isotopic enrichment of the meltwater through the late spring rainfalls (Unnikrishna et al., 2002). Several authors (Feng et





al., 2002; Hashimoto et al., 2002; Unnikrishna et al., 2002) reported a typical difference of $\delta^{18}O$ around 2‰ between solid snow and liquid water in snow which is mostly caused by the isotopic fractionation. Feng et al. (2002) reported that a difference of 1 ‰ in $\delta^{18}O$ is equivalent of 8 ‰ change in $\delta^2H$. Although this discrepancy can lead to some uncertainties in hydrograph separation, only little work has addressed the effects of the time-

variant isotopic content of the non-rain water.

Juras et al. (2016) demonstrated in a feasibility study that they could quantify the contribution of rainwater in snowpack runoff during a sprinkling experiment using hydrograph separation techniques. However, their experiment was conducted with very high sprinkling intensities well beyond typical rain intensities. In this paper, we extend their study to investigate the propagation of liquid water through snowpack under conditions

representative of natural ROS events and for different types of snowpack. Our data analysis allows answering the following questions:

1. How much does rain water contribute to the total snowpack runoff during ROS?
2. What is the interaction between rain and ripe or cold snowpack?
3. How do initial snowpack conditions of cold and ripe snow influence liquid water transport in snow?

In addition, we present a new approach to deal with isotopic differences within the initial snowpack, and test it against standard procedures.

## 2 Material and methods

### 2.1 Study site

Four sprinkling experiments were carried out in the vicinity of Davos, Switzerland. Elevation of the

experimental sites ranged between 1850 and 2150 m a.s.l. Details of all sites and experiments are summarised in table 1. All sites were located in open flat terrain. The winter season 2014/2015 was characterized by below-average snowcover and above-average mean air temperatures. Davos climate has a subalpine character with mean air winter temperature of -2.18°C and cumulative winter precipitation of 371 mm (Nov - Apr).

Table 1

### 2.2 Experimental procedure

Four ROS experiments were conducted in this study. During each experiment deuterium enriched water was sprinkled on an isolated block of snow, consisting of natural and undisturbed snow of 1m² surface area. Each experiment was conducted within three subsequent days: The first day, an experimental snow block of natural snow was prepared. To inhibit lateral exchange of water the snow block was carefully cut out and isolated from

adjacent snow using 4 sheets of Ethafoam® of 2cm thickness. A metal tray was pushed through the bottom section of the snow block in a slight angle enabling to collect liquid water from the lowest corner. The tray featured a rim of 5cm height on three of the four sides. The outlet channel was then attached to the fourth side, but only after the tray had been pushed through the snow block. The outlet was connected to a tipping bucket gauge, which also served to sample water for the laboratory analysis. The rainfall simulator was then placed





above the snow block with wind protection cover (Fig. 1) rolled up to ensure ambient thermal conditions. Even if mechanical and thermal disturbances were kept to a minimum the setup was allowed to settle over night before sprinkling experiments commenced the next day.

Figure 1

5 During the second day, the actual sprinkling onto the snow block was performed. Pre-experimental snow properties were measured in undisturbed snow within a few meters from the experiment at the time that the sprinkling started. We recorded vertical profiles of snow temperature, liquid water content (LWC), grain size and density. LWC was measured using a "Denoth meter" (Denoth, 1994). In addition snow samples were taken to analyse the $\delta^2H$ content. Snowpack runoff was recorded from two hours before the first sprinkling burst till five

10 hours after the last sprinkling burst. The snowpack runoff was further sampled for $\delta^2H$ content during the entire experiment. The sampling interval varied according to the snowpack runoff rate, ranging from one minute during the peak flow to 20 minutes during periods with marginal flow only. During the sprinkling, the wind protection cover was put in place to enable spatially homogenous sprinkling results. The cover was shortly opened during non-rain period to prevent possible accumulation of warm air. On day 3, approximately 20 hours after the last

sprinkling burst, post-experimental snow properties were measured analogously to day 2, with the exception that the sampling was conducted within the snow block that was sprinkled. Again, snow samples were taken to determine how much sprinkling water remained in the snowpack.

## 2.3 Rainfall simulation and monitoring

An enhanced version of the rainfall simulator described in Juras et al. (2013, 2016) was designed to achieve rain

intensities close to observations during natural ROS (Osterhuber, 1999; Rössler et al., 2014; Würzer et al., 2016a). The new device was equipped by a nozzle Lechler 460.368.30.CA which was precisely calibrated in the laboratory and again on site. The nozzle was placed 160 cm above the snow cover ensuring a spatially uniform rainwater distribution for the inner $1m^2$ of the sprinkling area, i.e. over the snow block.

During each experiment about 41 mm of deuterium enriched water was sprinkled in four sprinkling periods of 30

25 min each, separated by 30 min non-sprinkling periods, resulting in a mean rainfall intensity of 10.25 mm per hour, per burst respectively.

The deuterium content is expressed as a difference relative to Vienna Standard Mean Ocean Water (V-SMOW). For the purposes of an efficient hydrograph separation, tap water was enriched with deuterium to reach a difference of at least $\delta^2H$ = 60 ‰ V-SMOW between the snowpack and the sprinkling water. Sprinkling water

concentration ranged between $\delta^2H$ –23.11 to +22.61 ‰ V-SMOW and initial snowmelt deuterium concentration ranged between $\delta^2H$ –132.47 to -88.64 ‰ V-SMOW. The barrels containing the enriched sprinkling water were buried into snow to cool down the water temperature. The mean rain water temperature after pumping varied between 4.3 – 7.5°C (measured over the snow), which is considered representative of temperatures during natural rain on snow events in the area.





### 2.4 Sampling and laboratory analysis

Water samples collected during the experiments were stored in 10 or 20 ml plastic bottles. To minimize isotopic fractionation, air gaps in the samples were avoided and samples were subsequently frozen until the laboratory analysis. Snow samples were taken along three vertical profiles at 10 cm spacing before and after each experiment. Additionally, three samples of the entire snow profile were taken at the same time. All snow samples were melted at room temperature, filled into 10 ml plastic bottles and frozen until the laboratory analysis.

The analysis were carried out using a laser spectroscopy by LGR Inc. LWIA v2 facility of the Czech Technical University in Prague (Penna et al., 2010). Standard deviation of the results is $\delta^2$H 0.58 ‰ V-SMOW and 95 % confidence interval is $\delta^2$H 0.33 ‰ V-SMOW.

### 2.5 Data interpretation

The hydrograph separation technique was used to separate rainwater from the non-rain water in the total runoff:

$$Q_{total}(t) \times c_{total}(t) = Q_{rain}(t) \times c_{rain} + Q_{non-rain}(t) \times c_{non-rain}(t) \qquad (1)$$

$$Q_{total}(t) = Q_{rain}(t) + Q_{non-rain}(t), \qquad (2)$$

where $Q$ [mm·min$^{-1}$] is the flow rate, $c$ [‰ $\delta^2$H in V-SMOW] is the deuterium concentration and the subscripts *total*, *rain* and *non-rain* represent the total gauged snowpack runoff, the rainwater runoff and water originates from pre-experimental LWC and snowmelt respectively.

The non-rain water was considered as a mixture of two components pre-event liquid water content in the snowpack (pre-LWC) and the additional snowmelt water within the experimental snow block:

$$Q_{non-rain} = Q_{melt} + Q_{pre-LWC}. \qquad (3)$$

$Q_{melt}$ represents additional melt water produced during the experiment and $Q_{pre-LWC}$ represents pre-experimental liquid water content in the snowpack. Since the isotopic content of the snowpack varies within the vertical profile we assume that the reference value of non-rain water is time variant. According to previous investigations (Juras et al., 2016), rainwater appears in the snowpack runoff only after a certain delay. We can therefore assume that at the beginning of runoff the non-rain water consisted mostly of pre-LWC water ($Q_{pre-LWC}$). After some time contribution of pre-LWC retreats and additional melt water ($Q_{melt}$) starts to dominate within the non-rain runoff water volume. This water originating instantly from the solid phase has different isotopic content compared to pre-LWC (Feng, 2002; Hashimoto et al., 2002; Unnikrishna et al., 2002). The partitioning of the non-rain water in the snowpack ($c_{non-rain}$ in eq. 1) can be expressed as:

$$c_{non-rain} = \tan^{-1}\left(\frac{\frac{(T-t)\cdot 20\pi}{S}}{\pi} + 0.5\right) \cdot (c_{solid} - c_{melt}) + c_{melt}, \qquad (4)$$

where $T$ is time vector, $t$ [min] is time hypothetically needed to release all pre-LWC water, $S$ is a dimensionless parameter governing the shape of the curve, $c_{solid}$ is deuterium content of solid phase of the entire pre-experimental snowpack, $c_{melt}$ is deuterium content of pre-experimental meltwater. Parameter $t$ was derived as the time when volume of non-rain water equalled pre-LWC (Fig. 2). The temporal smoothing parameter S was set to




an arbitrary value of 45, c.f. section 4.4 for a discussion on the sensitivity of alternative approaches regarding eq (4). All fitted parameters are listed in Table 2. An illustration of the mixing curve is displayed in Figure 2

Table 2

The isotopic value of the pre-LWC non-rain water was derived from the sampling of the pre-experiment melt outflow and the isotopic value of the additional melt was derived from the sample of entire snowpack. The isotopic value of the rainwater was derived from the sampling of the water in barrel. In view of the short duration of the experiment, we don't assume any fractionation between solid and liquid phase during the sprinkling. Rainwater storage in the snow cube was estimated as:

$$Q_{stored} = Q_{rain-} - Q_{rain-out}. \qquad (5)$$

We define the LWC deficit as the non-rain water contribution to the snowpack runoff that cannot be satisfied from the initial LWC storage. Hence values above zero indicate the minimal snowmelt that must have occurred to provide LWC for the snowpack runoff. The LWC deficit is calculated as a cumulative deficit from the water balance as:

$$LWC_{deficit}(t) = \max\left(\sum_0^t V_{non-rain} - LWC_{init}, 0\right), \qquad (6)$$

where $LWC_{init}$ refers to initial total LWC summarised in Table 3 $V_{non-rain}$ refers to the volume of non-rain water in the runoff. Hydrograph data were analysed for time lag and peak times of each hydrograph component (Fig. 3). We define rainwater time lag as a time when rainwater runoff rate reaches $0.01 \text{ mm·h}^{-1}$ (according to Eq. 1, 2). Total water time lag is defined as a time difference between onset of the rain and the first significant increase of total water runoff above the base flow (consisting of melt) (Eq. 2). Peak time is defined as a time difference between onset of the rain and the time of runoff maximum of each hydrograph component.

Uncertainties of rainwater runoff contribution were estimated from using the spread between individual samples from the vertical snow profiles at 10 % and 90 % percentiles.

Figure 3

## 3 Results

### 3.1 Snowpack changes

Table 3 shows an overview of the pre-experimental and post-experimental snowpack conditions. The three snow blocks in Ex 2-4 consisted of snow with similar conditions being isothermal, well ripened with bulk densities above $400 \text{ kg·m}^{-3}$ and contained considerable initial liquid water. These snowpack conditions are referred to in the text as "ripe snow". Pre-experimental snowpack conditions in Ex1 differed from the other three. Snow temperatures were mostly below the freezing point and the bulk density was around $250 \text{ kg·m}^{-3}$. Despite this, a small amount of pre-experimental LWC was found in the top 5 cm, where the snow temperature was around the freezing point (Ex1). Nevertheless, these snowpack conditions are referred to as "cold snow".




Unlike our expectations, the experiments resulted in greater density changes in ripe snow compared to the changes in cold snow. The total bulk density increased by between 17 and 54 kg·m$^{-3}$ in Ex 2-4 compared to a 4 kg·m$^{-3}$ increase only in Ex 1 (Table 3),. On the contrary, LWC increased in all experiments by very similar values of approx. 2 %.

Table 3

An increased deuterium content of snow, caused by the isotopically enriched sprinkling water, indicate additional storage of rain water. Our results showed a considerable increase in deuterium content (Table 4) only for ex. 2-4 (ripe snow conditions).  In comparison, Ex. 1 (cold snow) showed a more ambiguous picture, indicating that only little rainwater volume remained in the snow after the experiment; if at all, the deuterium

content even decreased slightly (by -0.88 ‰). Details of the deuterium content of the main components before and after the experiments are listed in Table 4.

Table 4

### 3.2 Snowpack runoff

All experiments showed a quick response in snowpack runoff within 10 min (Ex1) to 27 min (Ex4) after the start

of sprinkling (Fig. 4). However, the first significant increase of deuterium content (signalizing the appearance of the rainwater) was detected in the runoff somewhat later which indicates that rainwater initially pushed out the pre-event LWC and only then started to contribute to the runoff with some delay. Time lags and peak flow times of main hydrograph components are summarised in Table 5. The difference between rainwater time lag and total water time lag indicates the delay with which rainwater appears in the snowpack runoff relative to other source

of LWC. Interestingly, this delay was considerable in experiments 2-4 (at least 12 minutes), but only minor (6 minutes) in experiment 1 which was the only one conducted on cold snow.

Also the difference between total runoff and rain runoff demonstrate that water from other sources than rain such as pre-experimental LWC dominated snowpack runoff at the beginning of the sprinkling experiment. Again it is experiment 1 that deviates from the other by featuring a higher rain contribution in the total runoff already

during the first sprinkling periods (Fig. 4). Towards the end of the experiment (sprinkling period 4) rain contributed only 27 % in Ex4 but 82 % in Ex1.

The total water time lag was similar between the four sprinkling periods of each experiment, with the exception of Ex1 that featured a considerably longer time lag in the first sprinkling period compared to all subsequent periods, which may hint at the development of preferential flow paths early on during the experiment.

Figure 4

Table 5

### 3.3 Water balance

All experiments showed a negative snowpack mass balance (Table 6), which is characterized by cumulative total runoff (output) exceeding the cumulative rain input (Fig. 5). This required that additional melt occurred during





all experiments. Cumulative event runoff computed according to Eq. 1 and 2 consisted of between 22.0 % (Ex4) and 76.4 % (Ex1) of rainwater (Table 6, Fig. 5). The storage of rainwater was calculated according to Eq. 5 which revealed that averaged over the entire experiment the snowpack retained 21.6 % (Ex1) to 69.6 % (Ex4) of the original rainwater volume. However, the rainwater storage ratio varied over the course of the experiment.

After the first sprinkling period the ratio was always highest and decreased with subsequent sprinkling periods (Table 6), and even depleted almost completely towards the end of Ex1.

Figure 6

The pre-LWC represented an important source of non-rain water in the snowpack runoff, especially during the first sprinkling period. The LWC deficit for each sprinkling period is shown in table 6. For example, in Ex1 only

0.9 mm of pre-LWC was available (Table 3), but 4.1 mm of non-rain water appeared in the outflow after the first sprinkling period (Table 6), resulting in a LWC deficit of 3.17 mm that must have been satisfied by snowmelt. In contrast, the initial snowpack in Ex2-4 contained sufficient pre-LWC to fully explain the non-rain component to the runoff from the first sprinkling period. But also towards the end of these experiments some meltwater is required to explain the observed snowpack runoff.

Table 6

**4 Discussion**

**4.1 Rainwater interaction with the snowpack**

Samples from snowpack runoff at the beginning of the sprinkling experiment revealed, that the first water to exfiltrate from the snowpack originated from pre-LWC, and not from the rain. Only with a certain time lag did

rain start to appear in the runoff samples. Obviously, rainwater introduced to the snowpack pushed existing pre-LWC water out of the snow block during the onset of the runoff generation. First water samples taken from the runoff featured a similar deuterium content as the pre-LWC, we may thus assume that pre-LWC predominated in the non-rain water at the beginning of the experiment, but as the pre-LWC storage depleted meltwater took over. The process where rainwater shifted the pre-LWC out of the snow matrix has been described as piston flow

(Feng et al., 2001; Unnikrishna et al., 2002). The piston flow effect probably played a role not only at the beginning of runoff generation. Time shifts in peak flow times suggest that rainwater pushed non-rain water even beyond the initial phase, although the effect weakened over the course of the experiment (Table 5). A similar behaviour was also described in Juras et al. (2016).

Comparing the volume of retained rainwater within the first sprinkling period with the amount of released non-

rain water (Table 6) reveals that in all experiments the initial snowpack had liquid water deficiency. Available pore space in the snowpack was filled after beginning of the sprinkling, which also resulted in relatively little rainwater runoff during the first sprinkling period. The rainwater contribution however increased during subsequent sprinkling periods, as available storage capacity for liquid water depleted and pre-LWC water exfiltrated. During all experiments the ratio of rainwater in total snowpack runoff was well below 100 % at all

35     times (Fig. 4). This indicates that some rainwater is constantly retained in the snowpack (refrozen or as LWC) over the entire course of the sprinkling within both, cold as well as ripe snow.





Differences in the results from Ex1 relative to results from the other experiments demonstrated that the contribution of rainwater to the runoff is influenced by the initial snowpack conditions. Cold snowpack containing low pre-LWC volume allowed high contribution of rainwater to the runoff (Ex1). On the other hand, ripe snowpack with considerable pre-LWC volume showed stronger indication of piston flow, which resulted in mostly non-rain water to appear in early snowpack runoff. Adding rain, pre-LWC and additional melt resulted in total cumulative runoff volumes exceeding the cumulative rain input on average by 27 % for the experiment with ripe snow (Ex2-4). To the contrary, runoff from the cold snowpack exceeded rain input only by 3 %.

**4.2 Hydrological response of snowpack with different snow conditions on ROS**

Our results showed that rainwater was transported much faster in cold snow (Table 5) which indicated the presence of preferential flow (Section 3.2). On the other hand, experiment with ripe snow, resulted in a much slower transport of rain water and showed evidence of the matrix flow regime. These findings are in agreement with previous studies of Schneebeli (1995) and Würzer et al. (2016b), but see Colbeck (1975) who reported long time lags to be typical for rain on cold snowpacks. Preferential flow is mostly observed in layered snow, where microstructural transitions can be found in the density profile (Hirashima et al., 2014; Techel et al., 2008). During preferential flow the wetting front is disaggregated into many smaller flow fingers, within which the hydraulic conductivity can be very high (Waldner et al., 2004) allowing water to be transported faster.

The hydraulic conductivity is connected to the intrinsic permeability of snow (Calonne et al., 2012) and varies with the snow grain size and density (Hirashima et al., 2014). The intrinsic permeability increases as the snow density decreases (Calonne et al., 2012), which is in agreement with our results. The snow in Ex1 was characterized by a low density and therefore supported faster generation of snowpack runoff compared to Ex2-Ex4. On the other hand, ripe snow typically features rounded snow grains and initial saturation which are associated with higher hydraulic conductivities. This opposing effect may have led to the findings of Colbeck (1975) cited above. In our experiments however, the distinctly lower density of the snow in Ex1 in combination with the occurrence of preferential flow seem to have prevailed other effects and caused a considerably faster transport of liquid water through the snowpack if compared to the experiments in ripe snow.

Ex1 aside, Ex2-4 showed similar initial snowpack conditions with the exception of snow depth (Table 3). This allowed to verify that rainwater time lags were expectedly sensitive to the transport distance. Time lags recorded during Ex4 were markedly longer than those recorded during Ex2-3, which supports a positive correlation between snow depth and water transport times as also noted by Wever et al. (2014).

**4.3 Internal mass exchange**

Our results provide an evidence of internal mass and energy exchange processes in the snowpack during the sprinkling experiments. Such processes represent refreezing of rainwater and generation of snowmelt (Avanzi et al., 2015; Wever et al., 2015), whereas mass has additionally been exchanged by the displacement of pre-LWC by rainwater.

After the first sprinkling period the cold snowpack in Ex1 released more non-rain water than can be explained by available pre-experimental LWC. The corresponding LWC deficit even increase over the course of the sprinkling





experiment (Table 6). This leads to the conclusion that snowmelt must have occurred as one of the processes involved in runoff generation. Further, rain water retained in the snowpack at the end of the experiment was larger than the final LWC which suggests that at the same time some rain water has been refrozen. Nevertheless, these processes may have been limited to comparably small amounts of water since the LWC deficit as well as the retained rain water volume were relatively small compared to the runoff volume. This conclusion is also supported by the small difference between the deuterium concentration of the snowpack before and after the experiment (Table 4).

Ex4, to the contrary, started with sufficient LWC to explain the runoff originating from non-rain water until sprinkling period 4. While snowmelt may or may not have happened during the entire experiment, it must at least have occurred during sprinkling period 4. But apparently pre-experimental LWC has dominated the runoff generated early on during the experiment (see discussion on piston flow regime). The same applies to Ex2 and 3, for which snowmelt was evidenced from at least sprinkling period 2 onwards. In all 3 experiments the deuterium concentration differed considerably in snow samples collected before and after the experiment. This suggests that mass exchange processes have had a larger turnover compared to Ex1.

### 4.4 Using a variable non-rain water reference in eq (4)

The deuterium concentration of pre-experimental melt water and samples taken from the entire snowpack profile differed within all experiments (Table 4). This is caused when snowmelt is not produced over the entire snow profile (Ex 1). Snowmelt prevails in the upper part of the snowpack. And indeed, the deuterium concentration of pre-experimental melt in Ex 1 was very close to values sampled from top level of the snow profile.

We can expect that the pre-experimental melt (sourcing from pre-LWC) is continuously depleted and meltwater is also concurrently produced from the snowpack with different isotopic concentration. This is why we introduced an enhanced approach of hydrograph separation between rainwater and non-rain water by allowing the non-rain water isotopic reference value to be variable in time. This method was compared to the more traditional approach (c.f. Juras et al., 2016) where a constant isotopic value is used from either pre-experimental meltwater or sampled from the entire snowpack. Also different parameters (t, S) in equation 4 were tested. Table 7 summarises rain water time lags, rainwater peak times and cumulative rainwater of all experiments that resulted from our sensitivity tests. While in general the differences between results from different approaches were small, notably different time lags resulted when using a constant isotopic value sampled from the entire snow column.

Especially in Ex1 when the isotopic value from the snowpack is used, the resulting rainwater time lag of 0 is unrealistic. While differences between the approaches are minor, using a time variant non-rainwater reference value seems to be a reasonable approach to arrive at more accurate estimations of rainwater time lags and outflow volumes.

Table 7





**5 Conclusion**

In this study we investigated liquid water transport behaviour through natural snow by means of sprinkling experiments. Using deuterated water enabled to disentangle the fate of rain water and initial liquid water content. Furthermore, the approach provided evidence of rain water refreezing and meltwater generation to occur together

over the course of the sprinkling experiments.

Interestingly, a sprinkling experiment on a cold snowpack resulted in markedly different water transport dynamics in comparison to experiments on melting snow. Snowpack runoff responded comparably quickly to the onset of sprinkling, and rainwater arrived in the runoff with a short delay only. The overall share of rainwater in the runoff was around 80 % indicating that internal mass exchange processes played a minor role. Data from

this experiment further suggested the development of preferential flow paths that allowed rainwater to propagate increasingly efficient through the snowpack as the sprinkling continued.

On the other hand, experiments conducted on wet isothermal snowpack, showed a different behaviour. Snowpack runoff was considerably delayed relative to the onset of the sprinkling, and consisted of initial liquid water content only. Rainwater appeared in the runoff only with further delay and with a relatively low share,

where the overall contribution of rainwater in the runoff did not exceed 50 %. At the same time, the total runoff volume exceeded rain input plus initial liquid water content which requires that additional water from snowmelt contributed to the runoff. Both findings demonstrate that internal mass exchange processes were important for runoff generation during rain on a melting snowpack.

**Data availability**

All data are available on request.

**Acknowledgement**

We would like to thank the scientific exchange program Sciex-NMS[ch] (project code 14.105), the Swiss Federal Office for the Environment FOEN and Internal Grant Agency of the Faculty of Environmental Sciences, CULS

Prague (project 20144254) and the Swiss National Foundation (Scopes) project Snow resources and the early prediction of hydrological drought in mountainous streams (SREPDROUGHT) IZ73ZO_152506/CZ for the funding of the project. Many thanks also belong to Timea Mareková and Pascal Egli for tremendous help and assistance during the field work. We also want to thank the SLF staff for technical support and Martin Šanda for isotope analysis.

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

40





Table 1 - Details of the experiments

| Site | Latitude | Longitude | Elevation | Label | Date |
|------|----------|-----------|-----------|-------|------|
| Sertig 1 | 46.7227267N | 9.8505897E | **1850 m** | Ex1 | *17. – 19.3 2015* |
| Sertig 2 | 46.7227856N | 9.8507236E | **1850 m** | Ex2 | *22. – 24.4. 2015* |
| Dischma | 46.7209731N | 9.9219625E | **2000 m** | Ex3 | *29.4 – 2.5. 2015* |
| Flüela | 46.7436736N | 9.9812761E | **2150 m** | Ex4 | *7. – 9.5. 2015* |

Table 2 – Parameters used in equation 4 for every single experiment.

| Experiment | t [min] | S [-] |
|------------|---------|-------|
| **Ex1** | 20 | 45 |
| **Ex2** | 95 | 45 |
| **Ex3** | 88 | 45 |
| **Ex4** | 215 | 45 |




Table 3 – Experimental snow block conditions before and after each experiment. Bulk density values were derived from the entire snow profile sample.

| Snow properties | Pre-experiment | | After-experiment | | Difference |
|---|---|---|---|---|---|
| | Mean | St. Dev. | Mean | St. Dev. | |
| *Ex1 – Sertig, Snow pits 17.-19.3.2015* | | | | | |
| Bulk density [kg.cm⁻³] | 247 | 4 | 251 | 8 | 4 |
| Total LWC [%] | 0.2 | 1.1 | 1.7 | 0.5 | 1.6 |
| Total LWC [mm] | 0.9 | 0.3 | 8.3 | 2.4 | 7.4 |
| Snow depth [cm] | 54.4 | 3.7 | 48.2 | 3.0 | -6.2 |
| Snow temperature [°C] | -1.0 | 0.6 | 0.0 | 0.0 | 1.0 |
| *Ex2 – Sertig, Snow pits 22.-24.4.2015* | | | | | |
| Bulk density [kg.cm⁻³] | 408 | 18 | 425 | 12 | 17 |
| Total LWC [%] | 3.7 | 0.1 | 5.3 | 0.7 | 1.6 |
| Total LWC [mm] | 11.0 | 0.3 | 13.9 | 1.1 | 2.8 |
| Snow depth [cm] | 29.7 | 2.2 | 25.8 | 2.1 | -3.9 |
| Snow temperature [°C] | 0.0 | 0.0 | 0.0 | 0.0 | 0.0 |
| *Ex3 – Dischma, Snow pits 29.-1.5.2015* | | | | | |
| Bulk density [kg.cm⁻³] | 403 | 33 | 457 | 14 | 54 |
| Total LWC [%] | 3.8 | 0.3 | 6.3 | 0.1 | 2.6 |
| Total LWC [mm] | 10.6 | 0.8 | 16.9 | 0.3 | 6.3 |
| Snow depth [cm] | 28.1 | 2.5 | 26.6 | 2.1 | -1.6 |
| Snow temperature [°C] | 0.0 | 0.0 | 0.0 | 0.0 | 0.0 |
| *Ex4 – Fluela, Snow pits 6.-8.5.2015* | | | | | |
| Bulk density [kg.cm⁻³] | 477 | 21 | 495 | 9 | 18 |
| Total LWC [%] | 3.5 | 0.5 | 5.6 | 0.3 | 2.1 |
| Total LWC [mm] | 28.7 | 4.3 | 45.8 | 3.7 | 17.1 |
| Snow depth [cm] | 88.4 | 2.1 | 81.6 | 2.4 | -6.8 |
| Snow temperature [°C] | 0.0 | 0.0 | 0.0 | 0.0 | 0.0 |

Table 4 – Overview of deuterium concentration changes within each experiment. Reference values were used in eq. 1 and 4 for hydrograph separation. Snow samples were taken by extracting a vertical core from the entire snow profile.

| | Pre-experimental reference value | | | Reference value after experiment | Difference |
|---|---|---|---|---|---|
| | *Rainwater* | *Melt water* | *Snow sample* | *Snow sample* | *Snow sample* |
| *Ex1* | -23.11 | -88.64 | -138.88 | -139.76 | -0.88 |
| *Ex2* | -5.60 | -123.49 | -120.41 | -116.32 | 4.09 |
| *Ex3* | 22.61 | -132.47 | -122.00 | -105.84 | 16.16 |
| *Ex4* | -13.16 | -118.66 | -127.48 | -116.22 | 11.26 |





Table 5 – Hydrograph analysis of different artificial rain-on-snow event.

| Sprinkling period | Time lag total [min] | Time lag rain [min] | Rainwater velocity [cm.min$^{-1}$] | Peak time total [min] | Peak time rain [min] |
|---|---|---|---|---|---|
| *Ex1 - Sertig 17.-19.3.2015 - snow depth = 54.4 cm* | | | | | |
| 1 | 10 | 16 | 3.40 | 27 | 33 |
| 2 | 4 | 4 | 13.60 | 22 | 27 |
| 3 | 4 | 4 | 13.60 | 20 | 27 |
| 4 | 5 | 5 | 10.88 | 25 | 25 |
| *Ex2 - Sertig 22.-24.4.2015 - snow depth = 29.7 cm* | | | | | |
| 1 | 15 | 27 | 1.10 | 35 | 40 |
| 2 | 13 | 13 | 2.28 | 31 | 36 |
| 3 | 17 | 17 | 1.75 | 28 | 10 |
| 4 | 13 | 14 | 2.12 | 30 | 10 |
| *Ex3 - Dischma 29.4.-1.5.2015 - snow depth = 29 cm* | | | | | |
| 1 | 13 | 26 | 1.08 | 33 | 36 |
| 2 | 9 | 9 | 3.12 | 29 | 34 |
| 3 | 11 | 11 | 2.55 | 28 | 31 |
| 4 | 9 | 9 | 3.12 | 27 | 27 |
| *Ex4 - Flüela 6.-8.5.2015 - snow depth = 88.4 cm* | | | | | |
| 1 | 27 | $\infty$* | na* | 50 | na* |
| 2 | 27 | 27 | 3.27 | 47 | 49 |
| 3 | 27 | 27 | 3.27 | 46 | 53 |
| 4 | 32 | 32 | 2.76 | 47 | 51 |

* rainwater was not recorded in response to the first sprinkling burst




Table 6 – Water balance computed from every outflow peak of the four experiments.

| Sprinkling period | Input [mm] | LWC deficit [mm] | Total out [mm] | Rain out [mm] | Rain out [%] | Non-Rain out [mm] | Volume rain stored [mm] | Volume Rain Stored [%] |
|---|---|---|---|---|---|---|---|---|
| *Ex1 - Sertig 17.-19.3.2015* | | | | | | | | |
| 1 | 10.39 | 3.17 | 8.14 | 4.04 | 49.65 | 4.10 | 6.35 | 61.10 |
| 2 | 10.39 | 5.36 | 11.48 | 9.29 | 80.95 | 2.19 | 1.10 | 10.56 |
| 3 | 10.39 | 6.87 | 10.52 | 9.01 | 85.62 | 1.51 | 1.38 | 13.31 |
| 4 | 10.39 | 9.15 | 12.53 | 10.26 | 81.85 | 2.27 | 0.13 | 1.29 |
| **Total** | **41.56** | | **42.67** | **32.60** | **76.40** | **10.07** | **8.96** | **21.56** |
| *Ex2 - Sertig 22.-24.4.2015* | | | | | | | | |
| 1 | 10.13 | 0 | 8.98 | 1.76 | 19.63 | 7.22 | 8.37 | 82.60 |
| 2 | 10.13 | 4.66 | 14.00 | 5.57 | 39.76 | 8.43 | 4.56 | 45.04 |
| 3 | 10.13 | 11.55 | 11.49 | 4.60 | 40.04 | 6.89 | 5.53 | 54.59 |
| 4 | 10.13 | 24.76 | 20.02 | 6.81 | 34.03 | 13.21 | 3.32 | 32.75 |
| **Total** | **40.52** | | **54.49** | **18.74** | **34.40** | **35.75** | **21.78** | **53.74** |
| *Ex3 - Dischma 29.4.-1.5.2015* | | | | | | | | |
| 1 | 10.39 | 0 | 7.20 | 1.58 | 21.89 | 5.62 | 8.81 | 84.83 |
| 2 | 10.39 | 0.25 | 10.44 | 5.14 | 49.21 | 5.30 | 5.25 | 50.55 |
| 3 | 10.39 | 4.98 | 11.14 | 6.41 | 57.55 | 4.73 | 3.98 | 38.30 |
| 4 | 10.39 | 11.55 | 16.22 | 9.64 | 59.46 | 6.58 | 0.75 | 7.17 |
| **Total** | **41.56** | | **45.00** | **22.77** | **50.60** | **22.23** | **14.25** | **45.21** |
| *Ex4 – Flüela 6.-8.5.2015* | | | | | | | | |
| 1 | 10.39 | 0 | 4.62 | 0.00 | 0.00 | 4.62 | 10.39 | 100.00 |
| 2 | 10.39 | 0 | 12.38 | 1.89 | 15.28 | 10.49 | 8.50 | 81.79 |
| 3 | 10.39 | 0 | 12.08 | 3.16 | 26.14 | 8.92 | 7.23 | 69.61 |
| 4 | 10.39 | 16.13 | 28.40 | 7.60 | 26.75 | 20.80 | 2.79 | 26.87 |
| **Total** | **41.56** | | **57.48** | **12.65** | **22.00** | **44.83** | **28.91** | **69.57** |




Table 7 – Different methods for estimation of reference non-rain water isotopic value were used in this table. 1. Constant value of a) entire snow sample, b) pre-experimental melt water and 2. Different parameters t, S in equation 4, where a) parameter used from Table 2, b) modified parameter from Table 2; t = t/2, S = S, c) modified parameter from Table 2; t = 2t, S = S, d) modified parameter from Table 2; t = t, S/2 = S, e) modified parameter from Table 2; t = t, S = 2S.

| | Non-rain reference isotopic source | Time lag rain [min] | | | | Peak time rain [min] | | | | Total rain output [min] | | | |
|---|---|---|---|---|---|---|---|---|---|---|---|---|---|
| | | Ex1 | Ex2 | Ex3 | Ex4 | Ex1 | Ex2 | Ex3 | Ex4 | Ex1 | Ex2 | Ex3 | Ex4 |
| 1 | a) Only snow | 0 | 29 | 31 | 39 | 30 | 42 | 38 | 62 | 34.2 | 18.2 | 21.6 | 16.2 |
| | b) Only melt | 16 | 27 | 26 | 87 | 33 | 40 | 36 | - | 28.1 | 19.1 | 23.2 | 12.7 |
| | a) Mixing - used | 16 | 27 | 26 | 87 | 33 | 40 | 36 | - | 32.6 | 18.8 | 22.8 | 12.8 |
| | b) Mixing - t/2 | 15 | 27 | 26 | 87 | 29 | 40 | 36 | - | 33.8 | 18.5 | 22.3 | 13.8 |
| 2 | c) Mixing - 2 t | 16 | 27 | 26 | 87 | 33 | 40 | 36 | - | 31.4 | 19.1 | 23.2 | 12.8 |
| | d) Mixing - S/2 | 16 | 27 | 26 | 87 | 33 | 40 | 36 | - | 32.5 | 18.8 | 22.8 | 12.8 |
| | e) Mixing - 2 S | 15 | 27 | 26 | 87 | 33 | 40 | 36 | - | 32.6 | 18.8 | 22.8 | 12.8 |





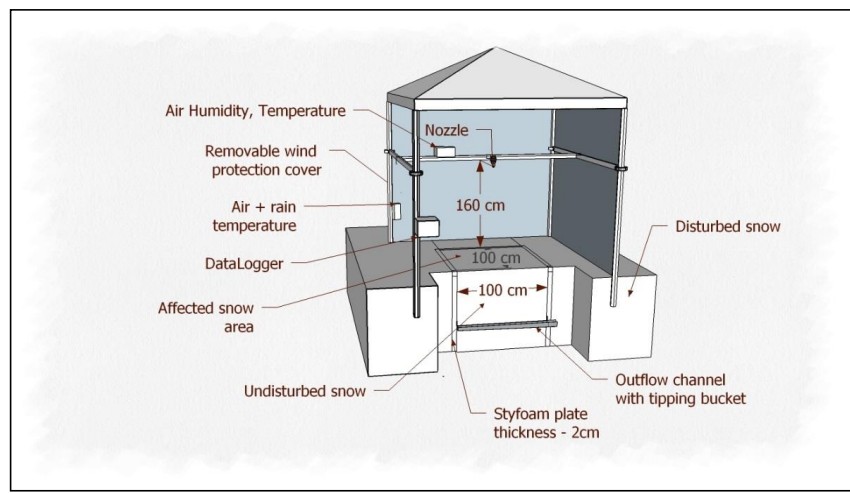

Figure 1 – Experimental setup of rainfall simulator.




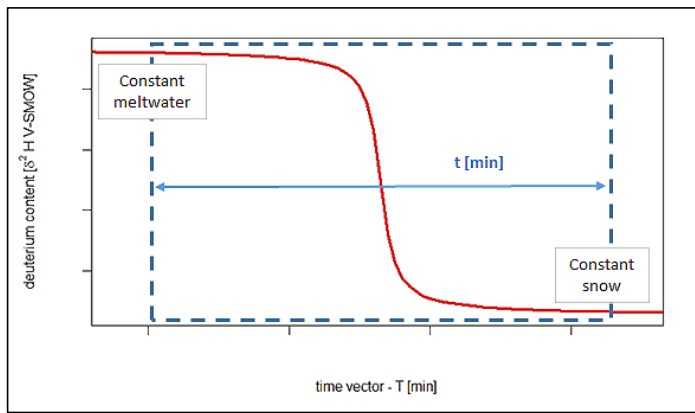

Figure 2 - Generalized mixing curve of non-rain water $c_{non\text{-}rain}(t)$ representing a transition from the deuterium concentration of pre-experimental LWC to a value which is influenced by additional melt.

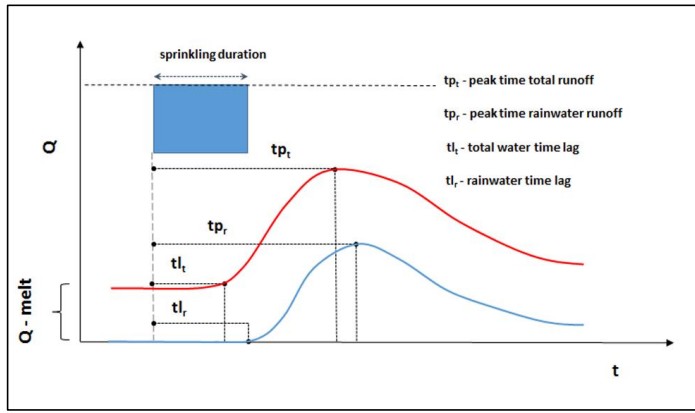

Figure 3 – Graphical definition of peak times and time lags.



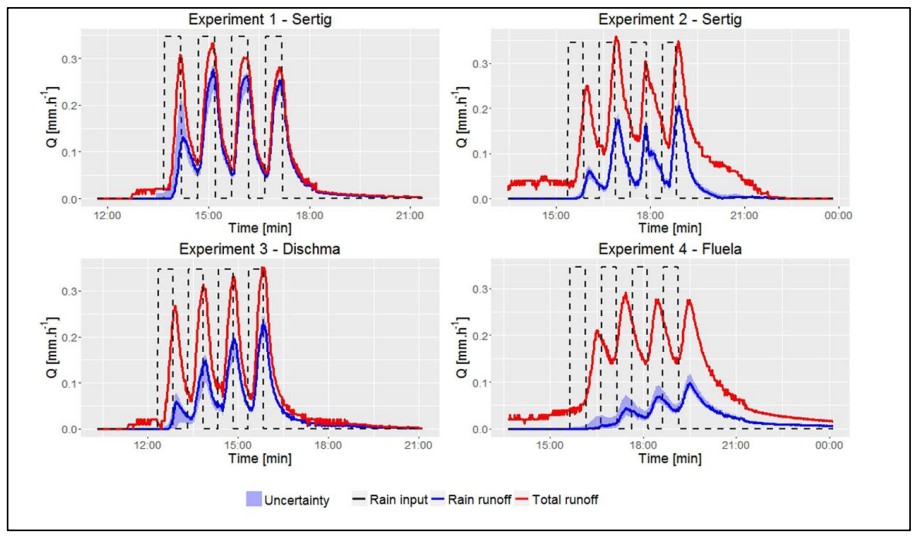

Figure 4 - Runoff from the experimental snow block during all artificial ROS.

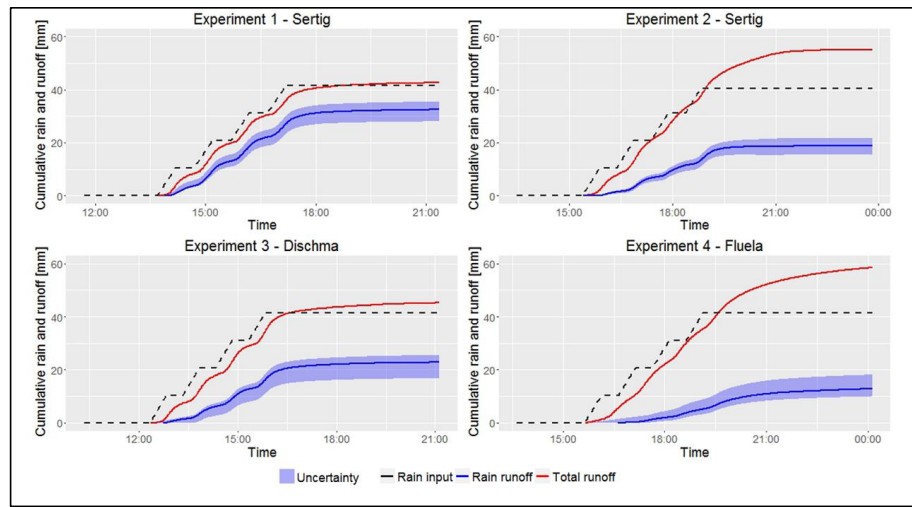

Figure 5 – Cumulative outflow from the investigated snow cube.