# Peer review of "Rainwater propagation through snowpack during rain-onsnow sprinkling experiments under different snow conditions"

_Hydrology and Earth System Sciences, 2016_

## Referee Comment (RC1) · Anonymous Referee #1 · 30 Dec 2016

The authors describe interesting sprinkling experiment, which were performed to study rain-on-snow events. They measured both outflow volumes and isotopic signals, which was possible due to the use of Deuterium enriched sprinkling water. They found that in cold/dry snow (unfortunately only one replicate) the outflow from the snow was faster both in terms of outflow reaction and rainwater travel times. While this finding could be expected with regard to the latter (i.e. rainwater travel times), the former (i.e. slower response of the outflow in wet/warm snow) seems counterintuitive. One explanation might be the development of preferential flow pathways, but without internal measurements/observations, this remains a bit speculative.

My major concern with this study is the not fully satisfactory explanation of the processes leading to the counterintuitive findings. Here I would find some more discussion/reasoning helpful, including a detailed discussion of potential errors, which could (not) explain this (especially since there was only one sprinkling experiment on cold/dry snow). Beyond this, my comments as listed below are rather minor:

Reading the manuscript, at some point I was confused by the four experiments and four rain pulses . . .. Probably it was me missing something, but this could perhaps also be described clearer.

The author present much of their observations in form of tables. The manuscript would become much more attractive if these results could be presented (also?) in form of figures.

While there obviously is a difference in scale, it would be useful to link the isotope studies in the present study to isotope studies at the catchment scale (e.g., Rodhe, 1981, Spring Flood Meltwater or Groundwater?)

P2L33: while melting snow and rain can have (and often have) a different isotopic composition, this difference is not a 'fact'

P3L3: What is meant by discrepancy here? Isn't this just the consequence of the GMWL?

P4L34: is there any evidence for these temperatures being representative?

Eq1: please avoid using x as multiplication sign

P5L15: delta values are no concentrations

Eq 4: where does this Eq and the tan in it come from?

P7L1: the sentence 'Unlike our expectations' sounds like discussion

P7L3: the location of 'only' seems strange, reformulate to clarify what is referred to by 'only'

While I am not a native speaker myself, I feel that there is some room for improvement with regard to the English. Among other things, the (not) use of 'the' seems not always correct and some sentences are a bit unclear to read (e.g. P2L19). The authors are also not fully consistent with the use of the tenses, and the tense used for reporting own work sometimes jumps between past and present.

---

## Referee Comment (RC2) · J. Garvelmann (Referee) · 4 Jan 2017

General comments:

The authors present a very interesting study about 4 sprinkling experiments with deuterium enriched water on natural snow covers with different initial conditions. The dynamics of snowpack outflow and the proportions of rainwater and melt water from the snowpack were analyzed using an hydrograph separation approach based on the deuterium signatures of the sprinkled rainwater, the snow cover and the runoff from the snowpack. The results of the study provide some very interesting insights into the dynamics of water flow within the snowpack during the artificial sprinkling experiments and are therefore highly relevant for the process knowledge of runoff generation dur-

ing ROS and consequently the improvement of hydrological models. The focus of the presented study is in the scope of HESS. I like the study very much. However, I recommend some revisions of the manuscript prior to a publication in HESS.

One of my main concerns about the submitted manuscript is the clear separation of experiment 1 from the other 3 experiments and the conclusions based on this one experiment having a cooler snow pack compared to the other experiments. From my point of view a snow pack described as "Snow temperature were mostly below the freezing point..." (page 6, lines 32 and 33) and the information from Table 3: Snow temperature -1.0°C with a standard deviation of 0.6°C can not be called a cold snow pack. The use of the term "cool" would be probably better. The results of experiment 1 are of course distinctly different from the other experiments. However, it is just one experiment and the other three experiments show also individual behavior. A clear separation and the conclusions are therefore critical. The authors should think about focusing on the individual behavior of each experiment. This would include a more detailed discussion on the shape of the observed runoff hydrographs in Figure 4 is lacking and would improve the study considerably. Why are the peaks of experiment 1 are decreasing from sprinkling period to sprinkling period, while the peaks in the other experiments tend to increase? Another point in that discussion may be the difference in the peak flows of total runoff and the rainwater fraction in total runoff. Furthermore, I highly motivate the authors to add a correlation analysis to further investigate the influences of snow pack properties (e.g. snow depth) on the observed hydrograph dynamics (e.g. lag times). This analysis would considerably improve the study and will provide further insight into the influences on different snow covers on the internal runoff generation.

The differences in total amounts of rainfall and runoff from the snowpack (page 9, lines 6 and 7 for example and Figure 5) are the reason why ROS events have the potential to generate more runoff than rainfall or snowmelt alone. Although the study in its current form is focused on the snow internal flow processes, please add a few more comments

and think about extending the discussion on that aspect of the study.

There is missing a few words on the scale issue (the experiment was performed on a square meter of snow. What can be expected on a larger scale, what literature is available on the runoff generation during ROS on larger scales?) as well a few words on the effects at the edges of the sprinkled snow block. Please provide also some discussion on the snowmelt energy balance during ROS and the influences this energy (that was certainly not available during the sprinkling experiments may have on the runoff generation within the snowpack. Furthermore, there is missing at least one figure in the results section showing the deuterium signatures during the sprinkling experiments.

Finally, I recommend removing or extending the analysis discussed in section 4.4. In its current form this part is too isolated from the rest of the study. However, the results of using a traditional hydrograph separation approach with snow or snowmelt isotope signature compared to the results with the presented approach would be highly interesting. The signature of the runoff observed prior to the actual sprinkling experiments (that is clearly visible in Figure 4 for all experiments) should be used, since Taylor et al. (2001 and 2002) recommend using the melt water stable isotope signature of the snowpack for an accurate isotope based hydrograph separation.

Specific comments:

I recommend the revision of the title of the presented study. Currently it is misleading, since the results of a number of artificial sprinkling experiments are shown and not the findings during a real ROS event.

In the introduction section there is missing more information about the previous modeling work (page 2, lines 9-12) as well as more details about the different flow concepts (page 2, lines 28-31).

There is missing some important literature (page 3, lines 1-5). Taylor et al. (2001 and

2002) point out that for hydrological applications (in their case isotope based hydrograph separation too) a correct representation of the snow pack is absolutely crucial. They recommend using the melt water stable isotope signature of the snowpack for that purpose.

From my point of view the use of "deuterium content" (page 4, line 9 for example) or "deuterium concentration" (page 4, line 30 for example) are not appropriate. Please use "deuterium signature" or "deuterium value" instead and correct throughout the whole manuscript.

Please provide a few more words about the melt runoff and its isotopic signature that was recorded already before the actual experiment started (page 4, line 9).

There are missing the information about the meteorological conditions prior and during the sprinkling experiments carried out.

Is c-solid (page 5, line 32) the average deuterium signature of the pre-experimental snowpack? Please specify. More information about the isotope signature (page 6, line 25) of the sampled snow profile would be very helpful.

Why was the deuterium value of the sprinkling water +22.61 per-mille VSMOW during experiment 3?

The paragraph on page 8 on lines 8 to 14 is very confuse and hardly understandable. Please revise for more clarity.

Do you refer to a certain experiment or to all experiments on Page 8, line 18?

Please mention clearly that the preferential flow may be due to the rapid development of fast flow paths in the snowpack when rainwater is infiltrating for more clarity (page 9, line 10).

Please provide a more comprehensive discussion on the hydrological response of the snow pack (section 4.2). Please provide more details about the Colbeck (1975) study.

Here some examples that may be relevant, among others of course, in order to improve the discussion on this aspect: Average liquid water holding capacity of 7% of an isothermal snowpack (Singh et al., 1997). Liquid water retention storage between 2% and 52% depending on snowpack conditions (Anderson, 1973). Kattelmann (1997): water outflow from 1 to 2 m snowpack between 4 and 6 hours after onset of rainfall.

The description of the methods are confuse at some points. Please provide the information on the methods used in the study in a very clear way.

There are mixed some results and discussion (page 7, lines 15-17 for example).

I recommend a careful proofreading of the final version of the revised manuscript prior to re-submission.

Technical notes:

Page 3, Line 23: average winter air temperature and mean annual winter precipitation for example.

Page 4, Line 30: Was is the deuterium signature of snow melt water or sampled solid snow later melted in the lab prior to analysis?

Page 5, Line 11: Date analysis would be the more adequate title of this section.

Page 6, Line 9: Please revise equation 5 (Q-rain-in).

Page 5, Line 19: "was" instead of "were".

Page 8, Line 20: rain water

Page 8, Line 30: deficit instead of deficiency

Page 8, Line 32: "...rainwater contribution, however, increased..."

Page 9, Line 8: The title of section 4.2 is confused. Please revise.

Page 9, Line 22: Please provide some literature at this point.

Page 9, Line 29: The point at the end of the sentence is missing.

Page 9, Lines 31 and 32: This sentence is too vague. Be careful with statements on the energy exchange processes within the snow pack based on the results of the study. Please revise this sentence.

Page 9, Line 32: Space too large.

Page 10, Line 3: ". . . refrozen or stored as liquid water in the snow pack." Please revise.

Page 10, Line 9: This sentence is too vague. Please revise.

Page 10, Line 11: Please provide information about were (section) the discussion on piston flow can be found in the manuscript.

Page 15, Table 1: Please revise the dates in table 1 (missing point, space).

Page 15, Table 2: Is this table really needed? Please check if the content can be included to the text or added to another table.

Page 16, Table 3: Please provide SWE of the snowpack in the table. Please provide the information of the structure analysis (grain size etc.) as mentioned in the methods section. Please revise unit of bulk density (kg*cm-3 instead of kg.cm-3). Please provide percentages to allow a better comparison of the different experiments.

Page 16, Table 4: Please provide units (per-mille VSMOW) in the table. Should it be different instead of difference in the header of the table? However, please revise the header text for more clarity.

Page 16, Table 3 and Table 3: Please thick about combining the two tables.

Page 17, Table 5: ". . .events" in the table caption. The peak times (10 min) for sprinkling period 3 and 4 in experiment 3 seem to be wrong. Please check.

Page 20, Figure 1: A real picture of the set-up of experiments would be nice to see.

Page 21, Figure 2: The influence of rainwater isotope signature is missing. Is this figure

really relevant and needed for the study?

References:

Taylor, S., Feng, X., Kirchner, J. W., Osterhuber, R., Klaue, B., and Renshaw, C. E.: Isotopic evolution of a seasonal snowpack and its melt, Water Resources Research, 37, 759-769, doi:10.1029/2000WR900341, 2001.

Taylor, S., Feng, X., Williams, M., and McNamara, J.: How isotopic fractionation of snowmelt affects hydrograph separation, Hydrological Processes, 16, 3683-3690, doi:10.1002/hyp.1232, 2002.

---

## Referee Comment (RC3) · J. Parajka (Referee) · 16 Jan 2017

General comments

The manuscript presents results of four experiments investigating rain percolation through the snowpack and snow melt runoff generation during rain-on-snow events. The rain was artificially generated by sprinkling deuterium enriched water. Contribution of rain and snowmelt on runoff generation was estimated by hydrograph separation technique. The results indicate that rain sprinkling on a colder snowpack had a different water transport dynamics compared to wet isothermal snowpack. Authors conclude that internal mass exchange is an important process for snowmelt runoff generation during rain-on-snow events.

[Figure]

This is an interesting study and worth to publish in HESS. However I also agree with the previous reviews that the clarity of the manuscript will benefit from some revision. I would suggest to make the formulation of title-objectives-results more consistent. The rainwater propagation/contribution/interaction does not have necessarily the same meaning and interpretation. Moreover I missed some more clear formulation of the research hypothesis. What is the main research question and how it can be accepted/rejected by performed experiments. Was there such a clear question prior to the setup of the experiment? Why and how were the four sites/dates selected? The last general comment is related to the discussion part – where it can be considered to add (I missed) some lessons learned section.

Overall I like the manuscript and enjoyed to reading it. I thus suggest some minor revision.

Specific comments

1) Abstract, l.14: the term "advanced hydrograph separation" is not clear here. Please consider to be more specific.

2) Eq.4. The form of the relationship is not clear. Some reference or more specific information would be useful.

3) Tables/Figures. Please consider to show some more main messages of the paper (presented now in Tables) in the form of figures.

4) Figure 4. Please consider to make the x axis longer, to show more clearly the timing. Perhaps the layout 1 column/4rows would be better.
* * *

---

## Author Comment (AC1) · 15 Mar 2017

**Reply to general comments of anonymous Referee #1:**

The authors describe interesting sprinkling experiment, which were performed to study rain-on-snow events. They measured both outflow volumes and isotopic signals, which was possible due to the use of Deuterium enriched sprinkling water. They found that in cold/dry snow (unfortunately only one replicate) the outflow from the snow was faster both in terms of outflow reaction and rainwater travel times. While this finding could be expected with regard to the latter (i.e. rainwater travel times), the former (i.e. slower response of the outflow in wet/warm snow) seems counterintuitive. One explanation might be the development of preferential flow pathways, but without internal measurements/observations, this remains a bit speculative.

My major concern with this study is the not fully satisfactory explanation of the processes leading to the counterintuitive findings. Here I would find some more discussion/reasoning helpful, including a detailed discussion of potential errors, which could (not) explain this (especially since there was only one sprinkling experiment on cold/dry snow).

*We would like to thank the referee for his/her helpful comments. We agree that finding a faster runoff response to the onset of sprinkling for cold snow may appear counterintuitive. While some discussion is already available we will expand the section in this regard specifically. We will also extend the discussion on differences between Ex2-4 (warm/wet snow) to address possible uncertainties. Note that we have further evidenced preferential flow paths in Ex. 1 by way of colour tracer, which we will document more clearly* (Würzer et al., 2016b). *Also the quick recession of runoff after the end of sprinkling hints at the presence of preferential flow during exp. 1.*

**Reply to specific comments of anonymous Referee #1:**

Beyond this, my comments as listed below are rather minor:

Reading the manuscript, at some point I was confused by the four experiments and four rain pulses … Probably it was me missing something, but this could perhaps also be described clearer. The author present much of their observations in form of tables. The manuscript would become much more attractive if these results could be presented (also?) in form of figures. While there obviously is a difference in scale, it would be useful to link the isotope studies in the present study to isotope studies at the catchment scale (e.g., Rodhe,1981, Spring Flood Meltwater or Groundwater?)

*We apologize if we failed to describe all four experiments as well as the respective four sprinkling periods in a clear way, obviously there is a need for improvement. We will try to better clarify our approach while revising the manuscript. Each of the four experiments consisted of four sprinkling periods lasting 30 minutes, separated by a 30 minutes break (See Fig. 4 in the manuscript). This approach was chosen to be able to investigate the temporal progression of response times to signals in the sprinkling input as the snowpack conditions changed over the course of the experiment. Additionally, note that rainfall intensity changes on sub hourly timescales can also be observed in nature.*

*We will expand out discussion on the possible implications of the study results on the catchment scale. We argue that some of the described mechanisms in the point scale have implications on the catchment scale, however processes such as overland flow or lateral flow in snow further add to the*

*complexity of runoff generation if concerned with the catchment scale. The presented hydrograph separation technique is transferable to larger scale, if the natural rain has constant isotopic signature (McDonnell et al., 1990). But linking to respective catchment studies is certainly beneficial for the discussion which we will implement as suggested.*

P2L33: while melting snow and rain can have (and often have) a different isotopic composition, this difference is not a 'fact'.

*We agree and will therefore rewrite the sentence to read: "Due to the often different isotopic signature of rain and snow, hydrograph separation can be applied to differentiate rainwater from the melt water in the total runoff from the snowpack."*

P3L3: What is meant by discrepancy here? Isn't this just the consequence of the GMWL?

*The sentence is redundant and will be deleted in the revised manuscript.*

P4L34: is there any evidence for these temperatures being representative?

*Assuming that rain temperatures are approximately equal to air temperatures, these temperatures are comparably warm but within range of observations. Unpublished data of rain temperatures during over 1000 natural ROS events evaluated in the context of (Würzer et al., 2016a), shown in Fig. 1 demonstrate that. Note that the direct effect of rain temperature on snowmelt is very small in comparison with other energy fluxes.*

[Figure]

*Fig. 1 – Representative mean air temperature during rain-on-snow events in Switzerland.*

Eq1: please avoid using x as multiplication sign

*We will use symbol "×" instead.*

P5L15: delta values are no concentrations

*We agree that the delta values are deficits to the V-SMOW standard deuterium concentration and will rewrite the sentence accordingly.*

Eq 4: where does this Eq and the tan in it come from?

*Equation 4 is a newly introduced formulation and represents an assumption on how the reference isotopic signature could change due to the piston flow effect (Fig.2). The tan function governs the shape of the gradual change of the deuterium reference value. It demonstrates that the reference value change is not a step function, but more likely S curve shape or reverse S curve shape, depending on initial snowmelt and snowpack signature.*

P7L1: the sentence 'Unlike our expectations' sounds like discussion

*This sentence will be reformulated in the revised manuscript.*

P7L3: the location of 'only' seems strange, reformulate to clarify what is referred to by 'only'.

*The sentence will be reformulated accordingly in the revised manuscript.*

While I am not a native speaker myself, I feel that there is some room for improvement with regard to the English. Among other things, the (not) use of 'the' seems not always correct and some sentences are a bit unclear to read (e.g. P2L19). The authors are also not fully consistent with the use of the tenses, and the tense used for reporting own work sometimes jumps between past and present.

*The English style and grammar will be carefully checked by a native speaker.*

**References**

*McDonnell, J. J., Bonell, M., Stewart, M. K. and Pearce, A. J.: Deuterium variations in storm rainfall: implications for stream hydrograph separation, Water Resour. Res., 26, 455–458, 1990.*

*Würzer, S., Jonas, T., Wever, N. and Lehning, M.: Influence of Initial Snowpack Properties on Runoff Formation during Rain-on-Snow Events, J. Hydrometeorol., 1801–1815, doi:10.1175/JHM-D-15-0181.1, 2016a.*

*Würzer, S., Wever, N., Juras, R., Lehning, M. and Jonas, T.: Modeling liquid water transport in snow under rain-on-snow conditions considering preferential flow, Hydrol. Earth Syst. Sci. Discuss., 18(August), 16488, doi:10.5194/hess-2016-351, 2016b.*

---

## Author Comment (AC2) · 15 Mar 2017

**Reply to general comments of J. Garvelmann:**

The authors present a very interesting study about 4 sprinkling experiments with deuterium enriched water on natural snow covers with different initial conditions. The dynamics of snowpack outflow and the proportions of rainwater and melt water from the snowpack were analysed using a hydrograph separation approach based on the deuterium signatures of the sprinkled rainwater, the snow cover and the runoff from the snowpack. The results of the study provide some very interesting insights into the dynamics of water flow within the snowpack during the artificial sprinkling experiments and are therefore highly relevant for the process knowledge of runoff generation during ROS and consequently the improvement of hydrological models. The focus of the presented study is in the scope of HESS. I like the study very much. However, I recommend some revisions of the manuscript prior to a publication in HESS.

One of my main concerns about the submitted manuscript is the clear separation of experiment 1 from the other 3 experiments and the conclusions based on this one experiment having a cooler snow pack compared to the other experiments. From my point of view a snow pack described as "Snow temperature were mostly below the freezing point: ::" (page 6, lines 32 and 33) and the information from Table 3: Snow temperature -1.0°C with a standard deviation of 0.6°C cannot be called a cold snow pack. The use of the term "cool" would be probably better. The results of experiment 1 are of course distinctly different from the other experiments. However, it is just one experiment and the other three experiments show also individual behavior. A clear separation and the conclusions are therefore critical. The authors should think about focusing on the individual behavior of each experiment. This would include a more detailed discussion on the shape of the observed runoff hydrographs in Figure 4 is lacking and would improve the study considerably. Why are the peaks of experiment 1 decreasing from sprinkling period to sprinkling period, while the peaks in the other experiments tend to increase? Another point in that discussion may be the difference in the peak flows of total runoff and the rainwater fraction in total runoff. Furthermore, I highly motivate the authors to add a correlation analysis to further investigate the influences of snow pack properties (e.g. snow depth) on the observed hydrograph dynamics (e.g. lag times). This analysis would considerably improve the study and will provide further insight into the influences on different snow covers on the internal runoff generation.

The differences in total amounts of rainfall and runoff from the snowpack (page 9, lines 6 and 7 for example and Figure 5) are the reason why ROS events have the potential to generate more runoff than rainfall or snowmelt alone. Although the study in its current form is focused on the snow internal flow processes, please add a few more comments and think about extending the discussion on that aspect of the study.

There is missing a few words on the scale issue (the experiment was performed on a square meter of snow. What can be expected on a larger scale, what literature is available on the runoff generation during ROS on larger scales?) as well a few words on the effects at the edges of the sprinkled snow block. Please provide also some discussion on the snowmelt energy balance during ROS and the influences this energy (that was certainly not available during the sprinkling experiments may have on the runoff generation within the snowpack. Furthermore, there is missing at least one figure in the results section showing the deuterium signatures during the sprinkling experiments.

Finally, I recommend removing or extending the analysis discussed in section 4.4. In its current form this part is too isolated from the rest of the study. However, the results of using a traditional hydrograph separation approach with snow or snowmelt isotope signature compared to the results with the presented approach would be highly interesting. The signature of the runoff observed prior to the actual sprinkling experiments (that is clearly visible in Figure 4 for all experiments) should be used, since Taylor et al. (2001 and 2002) recommend using the melt water stable isotope signature of the snowpack for an accurate isotope based hydrograph separation.

We would like thank Dr. Garvelmann for his detailed review. We appreciate his comments and suggestions. Please find our reply to all issues below.

The experiments were divided according to different snow properties, at first place the snow density and further the thermal state. Ex 1 was conducted during "mid-winter" condition, whereas Ex2-4 were conducted during melting period, when the snow density was already high. These differences are also reflected in the results. We agree that referring to the first experiment as "cold" experiment may not be ideal. We suggest using the term "non-ripe" instead, which describes the overall snow state better.

Thanks further as to the excellent suggestions for a more detailed discussion of the result, which will in particular deliver better insight in the variability between the four experiments.

We did a correlation analysis of initial snowpack properties (snow height, density, LWC) and the measures of runoff response (lag time, velocity) as suggested (Fig. 1). However, we are convinced that the number of experiments is not sufficient to inform such an analysis thoroughly; In particular given that one of the experiments is distinctively different from the others, the analysis will result in high but ill-founded correlations. Even if we appreciate the general idea of such an analysis, we suggest – for the above reasoning - not to present data as those exemplarily shown below.

Correlation among snow variables, n = 4

*Fig. 1 - Correlation analysis between snow properties (Initial LWC, Initial density, Snow depth) and runoff data (Flow velocity, Time lag)*

We will further expand the discussion on the possible implications of the study results on the catchment scale. We argue that some of the described mechanisms in the point scale have implications on the catchment scale, however processes such as overland flow or lateral flow in snow further add to the complexity of runoff generation if concerned with the catchment scale. The presented hydrograph separation technique is transferable to larger scale, if the natural rain has constant isotopic signature (McDonnell et al., 1990). The results will be further discussed and compared with earlier studies (Dincer et al., 1970; MacLean et al., 1995) which have addressed runoff composition within snow covered catchments.

Unfortunately the energy balance could not be meaningfully calculated because of missing short and longwave irradiation data inside the rainfall simulator. But we have prepared the plot of the deuterium signals as recommended which can be seen below in Fig. 2.

Fig. 2 – Suggestion of deuterium signature plot from all experiments.

We agree that chapter 4.4 should be extended since the new approach was introduced. The main message of this chapter is that using the pre-experimental meltwater deuterium content as a reference value for Eq. 1 only entail negligible differences in time lags. But noticeable difference may occur in the amount of rainwater in the total runoff. We will accentuate these findings in the chapter and also refer to Taylor et al.(2001, 2002) in the context of our results summarized in Tab. 7.

**Reply to specific comments of J. Garvelmann:**

I recommend the revision of the title of the presented study. Currently it is misleading, since the results of a number of artificial sprinkling experiments are shown and not the findings during a real ROS event.

We agree that the title could refer to the sprinkling experiments more specifically. We suggest changing the title to: "Rainwater propagation through snowpack during rain-on-snow sprinkling experiments under different snow conditions"

In the introduction section there is missing more information about the previous modelling work (page 2, lines 9-12) as well as more details about the different flow concepts (page 2, lines 28-31).

We will revise the introduction accordingly.

There is missing some important literature (page 3, lines 1-5). Taylor et al. (2001 and 2002) point out that for hydrological applications (in their case isotope based hydrograph separation too) a correct representation of the snow pack is absolutely crucial. They recommend using the melt water stable isotope signature of the snowpack for that purpose.

We will add this information and refer to the corresponding studies.

From my point of view the use of "deuterium content" (page 4, line 9 for example) or "deuterium concentration" (page 4, line 30 for example) are not appropriate. Please use "deuterium signature" or "deuterium value" instead and correct throughout the whole manuscript.

Thank you for this notice. We will consider using the term "deuterium signature" as suggested.

Please provide a few more words about the melt runoff and its isotopic signature that was recorded already before the actual experiment started (page 4, line 9).

We will add more information about the isotopic signature of pre-experimental meltwater.

There are missing the information about the meteorological conditions prior and during the sprinkling experiments carried out.

A short comment about the meteorological situation during the experiment will be included in the description of individual experiments.

Is c-solid (page 5, line 32) the average deuterium signature of the pre-experimental snowpack? Please specify. More information about the isotope signature (page 6, line 25) of the sampled snow profile would be very helpful.

Indeed, c-solid represents the average deuterium signature of the pre-experimental snowpack. More information will be added.

Why was the deuterium value of the sprinkling water +22.61 per-mille VSMOW during experiment 3?

It was important to maintain a minimum difference of 60 per-mile between sprinkling water and the solid snow. This difference was considered appropriate for a suitable rainwater separation. Setting of

maximal difference was not necessary, therefore it was not necessary to maintain the absolutely identical isotopic value of the sprinkling water for all four experiments.

The paragraph on page 8 on lines 8 to 14 is very confuse and hardly understandable. Please revise for more clarity.

We are sorry if this section caused any confusion. The paragraph will be revised for better reading.

Do you refer to a certain experiment or to all experiments on Page 8, line 18?

*Here, we refer to all experiments. The sentence will be revised for better clarity.*

Please mention clearly that the preferential flow may be due to the rapid development of fast flow paths in the snowpack when rainwater is infiltrating for more clarity (page 9, line 10). Please provide a more comprehensive discussion on the hydrological response of the snow pack (section 4.2). Please provide more details about the Colbeck (1975) study.

Here some examples that may be relevant, among others of course, in order to improve the discussion on this aspect: Average liquid water holding capacity of 7% of an isothermal snowpack (Singh et al., 1997). Liquid water retention storage between 2% and 52% depending on snowpack conditions (Anderson, 1973). Kattelmann (1997): water outflow from 1 to 2 m snowpack between 4 and 6 hours after onset of rainfall.

Thank you for these suggestions which will be considered to improve the discussion.

The description of the methods are confuse at some points. Please provide the information on the methods used in the study in a very clear way. There are mixed some results and discussion (page 7, lines 15-17 for example). I recommend a careful proofreading of the final version of the revised manuscript prior to re-submission.

We will follow the above suggestions and the manuscript will be revised carefully, including an English language check.

**Technical notes:**

Page 3, Line 23: average winter air temperature and mean annual winter precipitation for example. *The basic nomenclature will be unified.*

**Page 4, Line 30: Was the deuterium signature of snow melt water or sampled solid snow later melted in the lab prior to analysis?**

All frozen samples in the plastic bottles were melted in the lab prior to the analysis.

Page 5, Line 11: Date analysis would be the more adequate title of this section. *Thank you for the suggestion..*

**Page 6, Line 9: Please revise equation 5 (Q-rain-in).**

Thank you for the notice. The subscript Qrain-in will be revised.

**Page 5, Line 19: "was" instead of "were".**

We could not find any "were" in P5L19, but on P6L19. We use the plural form of the word "data" and the related plural verb form "were".

**Page 8, Line 20: rain water**

We would prefer to keep "rainwater" as it is through the entire manuscript.

**Page 8, Line 30: deficit instead of deficiency**

This will be corrected. Thank you.

Page 8, Line 32: ": : :rainwater contribution, however, increased ..." A comma will be added to the sentence in the revised manuscript.

Page 9, Line 8: The title of section 4.2 is confused. Please revise. *The section title will be revised.*

Page 9, Line 22: Please provide some literature at this point. Some relevant references will be added. E.g. (Fierz et al., 2009).

Page 9, Line 29: The point at the end of the sentence is missing. In our version of the manuscript, the punctuation is used correctly. This might be a technical problem with the pdf viewer?

Page 9, Lines 31 and 32: This sentence is too vague. Be careful with statements on the energy exchange processes within the snow pack based on the results of the study. Please revise this sentence.

The sentence will be revised. Thank you.

Page 9, Line 32: Space too large. *It will be corrected.*

Page 10, Line 3: "... refrozen or stored as liquid water in the snow pack." Please revise. *The sentence will be revised.*

Page 10, Line 9: This sentence is too vague. Please revise. *The sentence will be revised.*

Page 10, Line 11: Please provide information about were (section) the discussion on piston flow can be found in the manuscript. *This information can be found in chapter 4.1.*

Page 15, Table 1: Please revise the dates in table 1 (missing point, space). *The dates in table 1 will be revised.*

Page 15, Table 2: Is this table really needed? Please check if the content can be included to the text or added to another table.

The use of the table 2 will be once more considered.

Page 16, Table 3: Please provide SWE of the snowpack in the table. Please provide the information of the structure analysis (grain size etc.) as mentioned in the methods section. Please revise unit of bulk density (kg\*cm-3 instead of kg.cm-3). Please provide percentages to allow a better comparison of the different experiments.

We think that SWE would provide a redundant information, because density and snow depth are already in the table. We do not have comprehensive information about the grain size from all experiments. The density unit will be corrected –  $(kg m^{-3})$ .

Page 16, Table 4: Please provide units (per-mille VSMOW) in the table. Should it be different instead of difference in the header of the table? However, please revise the header text for more clarity.

The units will be provided. We prefer using "difference" in the header as a result of subtraction. The header text will be revised.

**Page 16, Table 3 and Table 3: Please thick about combining the two tables.**

This comment was probably meant to combine Tables 5 and Table 6. We think that a combination of these tables would not be beneficial for the paper, because it would contain too much information. In the current manuscript Table 5 represents the results of hydrograph times and water velocity. On the other hand Table 6 represents results of water volumes within the hydrographs. We will consider a suitable combination without losing the information clarity.

**Page 17, Table 5: ": : :events" in the table caption. The peak times (10 min) for sprinkling period 3 and 4 in experiment 3 seem to be wrong. Please check.**

Plural will be added to the table caption. The peak times were checked and confirmed as correct.

**Page 20, Figure 1: A real picture of the set-up of experiments would be nice to see.**

Unfortunately, we do not have an appropriate real picture to add.

Page 21, Figure 2: The influence of rainwater isotope signature is missing. Is this figure really relevant and needed for the study?

The Figure represents the new hydrograph separation concept. However, it is just a graphical representation of formula 4 and therefore we will consider removing it.

**References:**

Dinçer, T., Payne, B. R., Florkowski, T., Martinec, J. and Tongiorgi, E.: Snowmelt runoff from measurements of tritium and oxygen-18, Water Resour. Res., 6(1), 110–124, doi:10.1029/WR006i001p00110, 1970.

*Fierz, C., Armstrong, R. L., Durand, Y., Etchevers, P., Greene, E., Mcclung, D. M., Nishimura, K., Satyawali, P. K. and Sokratov, S. A.: The International Classification for Seasonal Snow on the Ground, 1st ed., IACS, Paris., 2009.*

MacLean, R. A., English, M. C. and Schiff, S. L.: Hydrological and hydrochemical response of a small Canadian Shield catchment to late winter rain-on-snow events, Hydrol. Process., 9(April), 845–863, doi:10.1002/hyp.3360090803, 1995.

McDonnell, J. J., Bonell, M., Stewart, M. K. and Pearce, A.

---

## Author Comment (AC3) · 15 Mar 2017

**Reply to general comments of J. Parajka:**

The manuscript presents results of four experiments investigating rain percolation through the snowpack and snow melt runoff generation during rain-on-snow events. The rain was artificially generated by sprinkling deuterium enriched water. Contribution of rain and snowmelt on runoff generation was estimated by hydrograph separation technique. The results indicate that rain sprinkling on a colder snowpack had a different water transport dynamics compared to wet isothermal snowpack. Authors conclude that internal mass exchange is an important process for snowmelt runoff generation during rain-on-snow events.

This is an interesting study and worth to publish in HESS. However I also agree with the previous reviews that the clarity of the manuscript will benefit from some revision. I would suggest to make the formulation of title-objectives-results more consistent. The rainwater propagation/contribution/interaction does not have necessarily the same meaning and interpretation. Moreover I missed some more clear formulation of the research hypothesis. What is the main research question and how it can be accepted/rejected by performed experiments. Was there such a clear question prior to the setup of the experiment? Why and how were the four sites/dates selected? The last general comment is related to the discussion part – where it can be considered to add (I missed) some lessons learned section.

*We would like to thank Dr. Parajka for his helpful comment.*

*We will carefully reassess the uses of terms such as "propagation", "contribution" and "interaction". "Propagation" is used for describing the transport process of liquid water within the snowpack. "Contribution" refers to the volume of runoff originated in rainwater or meltwater, whereas "interaction" refers to melt/refreeze and displacement processes involving rainwater, liquid water content and ice matrix. Nevertheless, we suggest new title of the paper "Rainwater propagation through snowpack during rain-on-snow sprinkling experiments under different snow conditions"*

*We intentionally avoided using research hypothesis, as the number of experiments is too small to allow for significance tests needed to accept/reject hypothesis. The main research idea is formulated in P3L9 and is further detailed in three research questions P3L12-14. All questions were formulated prior to the experiments and the experiments were designed according to these questions.*

*The experimental sites were selected to guarantee sufficient snow depth to conduct the experiments towards the end of snow season. Additionally, reachability/safety reasons/technical feasibility for transport of the equipment limited the choice of possible sites.*

*We will thoroughly consider your above points when revising the manuscript.*

Overall I like the manuscript and enjoyed to reading it. I thus suggest some minor revision.

**Reply to specific comments of J. Parajka:**

1) Abstract, l.14: the term "advanced hydrograph separation" is not clear here. Please consider to be more specific.

*The term "advanced" addresses that the approach employed in this paper additionally accounts for temporal changes in the isotopic signature of the reference values. We will specify this in the revised manuscript.*

2) Eq.4. The form of the relationship is not clear. Some reference or more specific information would be useful.

*Equation 4 is a newly presented formulation and represents an assumption on how the reference isotopic signature could change during the piston flow effect (Fig. 2 in the manuscript). The tan function governs the shape of the gradual change of the deuterium reference value. It demonstrates that the reference value change is not a step function, but more likely S curve shape or reverse S curve shape (It depends on initial snowmelt and snowpack signature.).*

3) Tables/Figures. Please consider to show some more main messages of the paper (presented now in Tables) in the form of figures.

*We initially planned to present data in form of figures, but it was difficult to display the same amount of information as in tabular form. Nevertheless, we will consider the reviewer's suggestion to display at least the main findings from tables 5 and 6 in an additional figure.*

4) Figure 4. Please consider to make the x axis longer, to show more clearly the timing. Perhaps the layout 1 column/4rows would be better.

*Thanks for this comment which we will implement as suggested (See Fig. 1).*

[Figure]

*Fig. 1 – An updated plot of experimental runoffs.*

---

## Author Response (AR1)

Dear Prof. Dr. Markus Weiler,

We would like to thank you and the three reviewers for the inspiring comments, which were really helpful for improving the manuscript. Please follow our discussion regarding the reviewers' comments in this document. We provide both, a revised manuscript and a version

5 with all changes marked. Please note that in our replies to the reviewer's comments we refer to the line numbers in the original manuscript, whereas references to changes in the revised manuscript refer to the line numbers in the manuscript version with all changes marked.

The individual replies below consist of the original reviewer's comment, our reaction that

10 has emerged from the on-line review *(italic),* and the actual changes and/or adjustments performed in this final manuscript to address the original comments.

Thank you

**15 Yours sincerely**

Roman Juras (on behalf of the authors)

**Reply to general comments of anonymous Referee #1:**

The authors describe interesting sprinkling experiment, which were performed to study rain-on-snow events. They measured both outflow volumes and isotopic signals, which was possible due to the use of Deuterium enriched sprinkling water. They found that in cold/dry snow (unfortunately only one

- 5 replicate) the outflow from the snow was faster both in terms of outflow reaction and rainwater travel times. While this finding could be expected with regard to the latter (i.e. rainwater travel times), the former (i.e. slower response of the outflow in wet/warm snow) seems counterintuitive. One explanation might be the development of preferential flow pathways, but without internal measurements/observations, this remains a bit speculative.
- 10 My major concern with this study is the not fully satisfactory explanation of the processes leading to the counterintuitive findings. Here I would find some more discussion/reasoning helpful, including a detailed discussion of potential errors, which could (not) explain this (especially since there was only one sprinkling experiment on cold/dry snow).

We would like to thank the referee for his/her helpful comments. We agree that finding a faster
runoff response to the onset of sprinkling for cold snow may appear counterintuitive. While some discussion was already available we have expanded the section in this regard specifically. We have also extended the discussion on differences between Ex2-4 (warm/wet snow) to address possible uncertainties. Note that we have further evidenced preferential flow paths in Ex. 1 by way of colour tracer, which we will document more clearly (Würzer et al., 2017). Also the quick recession of runoff

20 after the end of sprinkling hints at the presence of preferential flow during exp. 1.

Changes: We expanded Section 4.2, where the generation of preferential flow is now discussed in more detail. As a consequence, this section was also renamed to "Rainwater transport within different flow regimes".

**25 **Reply to specific comments of anonymous Referee #1:**

Beyond this, my comments as listed below are rather minor:

Reading the manuscript, at some point I was confused by the four experiments and four rain pulses ... Probably it was me missing something, but this could perhaps also be described clearer. The author present much of their observations in form of tables. The manuscript would become much more

30 attractive if these results could be presented (also?) in form of figures. While there obviously is a difference in scale, it would be useful to link the isotope studies in the present study to isotope studies at the catchment scale (e.g., Rodhe,1981, Spring Flood Meltwater or Groundwater?)

We apologize if we failed to describe all four experiments as well as the respective four sprinkling periods in a clear way, obviously there was a need for improvement. We have tried to better clarify

- 35 our approach while revising the manuscript. Each of the four experiments consisted of four sprinkling periods lasting 30 minutes, separated by a 30 minutes break (See Fig. 4 in the manuscript). This approach was chosen to be able to investigate the temporal progression of response times to signals in the sprinkling input as the snowpack conditions changed over the course of the experiment. Additionally, note that rainfall intensity changes on sub hourly timescales can also be observed in approach.
- 40 nature.

*Changes:* We rewrote the description of the experimental procedure and better clarified the sprinkling process (Section 2.3, Lines 20-24, Page 5).

We have further considered to present some results in form of figures and deleted one table (original Table 2) and added one figure (new Figure 4)

We have expanded our discussion on the possible implications of the study results on the catchment
scale. We argue that some of the described mechanisms in the point scale have implications on the catchment scale, however processes such as overland flow or lateral flow in snow further add to the complexity of runoff generation if concerned with the catchment scale. The presented hydrograph separation technique is transferable to larger scale, if the natural rain has constant isotopic signature (McDonnell et al., 1990). But linking to respective catchment studies is certainly beneficial for the
discussion which we will implement as suggested.

- Changes: We extended the discussion in Section 4.3, where the transferability of results between point scale and catchment scale is now discussed in more details. Furthermore, more relevant references were added. Please see Lines 13-21, Page 12.
- 15 P2L33: while melting snow and rain can have (and often have) a different isotopic composition, this difference is not a 'fact'.

We agree and have therefore rewritten the sentence to read: "Due to the often different isotopic signature of rain and snow, hydrograph separation can be applied to differentiate rainwater from the melt water in the total runoff from the snowpack."

**20 *Changes:* Lines 16-17, Page 3.**

P3L3: What is meant by discrepancy here? Isn't this just the consequence of the GMWL?

The sentence is redundant and has been deleted in the revised manuscript.

**Changes: The sentence was deleted.**

25

**P4L34: is there any evidence for these temperatures being representative?**

Assuming that rain temperatures are approximately equal to air temperatures, these temperatures are comparably warm but within range of observations. Unpublished data of rain temperatures during over 1000 natural ROS events evaluated in the context of (Würzer et al., 2016a), shown in

**30** Fig. a demonstrate that. Note that the direct effect of rain temperature on snowmelt is very small in comparison with other energy fluxes.

*Fig. a – Representative mean air temperature during rain-on-snow events in Switzerland.*

**Changes: No changes.**

Eq1: please avoid using x as multiplication sign

5 We now use the symbol "×" instead.

**Changes: (Updated Eq. 1.)**

**P5L15: delta values are no concentrations**

We agree that the delta values are deficits to the V-SMOW standard deuterium concentration and 10 have rewritten the sentence accordingly.

Changes: We changed the nomenclature, where only the term "deuterium signature" is used instead of terms such as "deuterium content" and "deuterium concentration" throughout the revised manuscript.

**15 Eq 4: where does this Eq and the tan in it come from?**

Equation 4 is a newly introduced formulation and represents an assumption on how the reference isotopic signature could change due to the piston flow effect (Fig.2). The tan function governs the shape of the gradual change of the deuterium reference value. It demonstrates that the reference value change is not a step function, but more likely S curve shape or reverse S curve shape, depending on initial snowmelt and snowpack signature.

20

**Changes: No changes.**

**P7L1: the sentence 'Unlike our expectations' sounds like discussion**

This sentence has been reformulated in the revised manuscript.

25 Changes: Lines 11-12, Page 8. P7L3: the location of 'only' seems strange, reformulate to clarify what is referred to by 'only'.

The sentence has been reformulated.

Changes: See Lines 31, Page 10.

- 5 While I am not a native speaker myself, I feel that there is some room for improvement with regard to the English. Among other things, the (not) use of 'the' seems not always correct and some sentences are a bit unclear to read (e.g. P2L19). The authors are also not fully consistent with the use of the tenses, and the tense used for reporting own work sometimes jumps between past and present.
- 10 The English style and grammar has been carefully checked by a native speaker.

**Changes:** (Throughout the entire manuscript)

**Reply to general comments of J. Parajka:**

The manuscript presents results of four experiments investigating rain percolation through the
 snowpack and snow melt runoff generation during rain-on-snow events. The rain was artificially
 generated by sprinkling deuterium enriched water. Contribution of rain and snowmelt on runoff
 generation was estimated by hydrograph separation technique. The results indicate that rain
 sprinkling on a colder snowpack had a different water transport dynamics compared to wet
 isothermal snowpack. Authors conclude that internal mass exchange is an important process for
 snowmelt runoff generation during rain-on-snow events.

This is an interesting study and worth to publish in HESS. However I also agree with the previous reviews that the clarity of the manuscript will benefit from some revision. I would suggest to make the formulation of title-objectives-results more consistent. The rainwater propagation/contribution/interaction does not have necessarily the same meaning and

- 25 interpretation. Moreover I missed some more clear formulation of the research hypothesis. What is the main research question and how it can be accepted/rejected by performed experiments. Was there such a clear question prior to the setup of the experiment? Why and how were the four sites/dates selected? The last general comment is related to the discussion part where it can be considered to add (I missed) some lessons learned section.
- 30 We would like to thank Dr. Parajka for his helpful comment.

We have carefully reassessed the uses of terms such as "propagation", "contribution" and "interaction". "Propagation" is used for describing the transport process of liquid water within the snowpack. "Contribution" refers to the volume of runoff originated in rainwater or meltwater, whereas "interaction" refers to melt/refreeze and displacement processes involving rainwater, liquid

35 water content and ice matrix. Nevertheless, we suggest a new title of the paper "Rainwater propagation through snowpack during rain-on-snow sprinkling experiments under different snow conditions"

Changes: The terms "propagation", "contribution" and "interaction" were revised throughout the entire manuscript. We further changed the title of the paper.

40

We intentionally avoided using research hypothesis, as the number of experiments is too small to allow for significance tests needed to accept/reject hypothesis. The main research idea is formulated in P3L9 and is further detailed in three research questions P3L12-14. All questions were formulated prior to the experiments and the experiments were designed according to these questions.

**5 Changes: The research question Nr. 2 was reformulated. See Lines 4-5, Page 4.**

The experimental sites were selected to guarantee sufficient snow depth to conduct the experiments towards the end of snow season. Additionally, reachability/safety reasons/technical feasibility for transport of the equipment limited the choice of possible sites.

10

20

25

30

*Changes:* We extended Section 4.4, where also text on the "lessons learned" has been added. See Lines 5-9, Page 13.

Overall I like the manuscript and enjoyed to reading it. I thus suggest some minor revision.

**15 Reply to specific comments of J. Parajka:**

1) Abstract, I.14: the term "advanced hydrograph separation" is not clear here. Please consider to be more specific.

The term "advanced" addresses that the approach employed in this paper additionally accounts for temporal changes in the isotopic signature of the reference values. We now specify this in the revised manuscript.

**Changes: The term "advanced" was replaced by "alternative". The alternative hydrograph separation is described by Eq. 4.**

2) Eq.4. The form of the relationship is not clear. Some reference or more specific information would be useful.

Equation 4 is a newly presented formulation and represents an assumption on how the reference isotopic signature could change during the piston flow effect (Fig. 2 in the manuscript). The tan function governs the shape of the gradual change of the deuterium reference value. It demonstrates that the reference value change is not a step function, but more likely S curve shape or reverse S curve shape (It depends on initial snowmelt and snowpack signature.).

**Changes: No changes.**

3) Tables/Figures. Please consider to show some more main messages of the paper (presented now in Tables) in the form of figures.

**35** We have considered to present some results in form of figures instead of tables. While we find tables to be an efficient way of presenting the type of results that originate from this study, we have deleted one table (original Table 2) and added one figure (new Figure 4).

**Changes: Deleted table 2, new figure 4.**

4) Figure 4. Please consider to make the x axis longer, to show more clearly the timing. Perhaps the layout 1 column/4rows would be better.

---

## Author Response (AR2)

Dear Prof. Dr. Markus Weiler,

We would like to thank you and reviewer Dr. Jakob Garvelmann for accepting most of our previous changes and improvements of the manuscript. We are also thankful for the additional comments. Please follow our discussion regarding the reviewer's comments in this document. We provide both,

5    a revised manuscript and a version with all changes marked. Please note that in our replies to the reviewer's comments we refer to the line numbers in the last version of revised manuscript, whereas references to changes in the revised manuscript refer to the line numbers in the new version of the manuscript without marked changes.

The individual replies below consist of the original reviewer's comment and our reaction with

10    the actual changes in bolt text.

15    Thank you

Yours sincerely

Roman Juras (on behalf of authors)

hess-2016-612: Rainwater propagation through snowpack during rain-on-snow events under different snow condition by Juras et al.

The manuscript about the results of 4 sprinkling experiments with deuterium enriched water on natural snow blocks has sufficiently improved during the review process. However, there are still some minor revisions that need to be realized prior to a publication in HESS.

Page 1, Line 19: The snowpack runoff was continuously recorded. The water was, however sampled in a certain temporal resolution. Please specify.

**Changes: The sentence was rewritten. See P1L20.**

Page 1, Abstract: Please mention that there was no proof for refreezing.

**Changes: This comment seems misplaced as we do not mentioned refreezing in the abstract at all. No changes.**

Page 1, Line33: The benchmark study on a large food in the Pacific North West (Marks et al., 1998) is missing here.

**Changes: The suggested benchmark study was added.**

Page 2, line 15ff: There is missing some information on studies and the mechanisms of refreezing in cold snow packs (formation of ice layers etc.)

**Changes: Additional information was added. See P2L29-33.**

Page 4, Line34: You would like to refer to Figure 5, right? However, this reference is not needed here from my point of view.

**Changes: Agreed, the figure reference was deleted.**

Page 5, Line 2: Please add some information on how and where the snowpack was sampled prior to the experiments.

**Changes: This information is available on P4L17 ff. No changes.**

Page 5, Line 10: I guess the samples after the experiments were collected at the study plot. The samples before were collected a few meters away as explained earlier in the text. Please specify here.

**Changes: Section 2.2 outlines the experimental procedure. The location of snow samples is mentioned there. Please read P4L17ff. Here focus is on the sampling itself.**

Page 5, Line 14: was instead of were

**Changes: The sentence was rewritten.**

Page 5, Line 14ff: Please proved a bit more info on the stable isotope analysis. About precision of the analysis. Where replicate house standards used and replicate samples measured etc. The last sentence of this section is confusing (I guess you are referring to the measurement accuracy.... Please clarify.

**Changes: The section was rewritten and extended. See P5L20-25.**

Page 6, Line 31: What do you mean with significant? Please quantify.

**Changes: A definition was added. P7L1-2.**

Page 6, Line 35+36: This is may it is important to know the location of the sampled profiles as suggested earlier.

**Changes: Please see above.**

The results section is fairly short now. But I think the most important points are picked out. Please think about adding some additional information at some points.

**Changes: The result section was extended. See sections 3.1 and 3.2.**

There is definitely missing an explanation of Figure 4 in the results section. The isotope dynamic of experiment 2 is obviously different from the other 3 experiments. Please provide a detailed description about this.

**Changes: We have added more explanation as to the data shown in Figure 4 in Section 4.3. See P11L10-13.**

You are describing the snow covers in experiment 2-4 as ripe melting snow packs. The snow in experiment 1 as non-ripe in contrast. Figure 5 clearly shows that there was a little bit of runoff prior to the onset of the sprinkling. Which means that there was already some snow melt going on. What are the reasons? There is missing an explanation in the text.

**Changes: Please refer to changes in Section 3.1 (P7L14-17) and Section 4.2 (P10L14ff).**

Furthermore it would be helpful to provide the results of the grain size analysis you carried out of the snowpack of the different experiments.

**Changes: We removed mentioning of the grain size analysis from the manuscript (Section 2.2).**

Page 7, Line 7: "Snow temperature were mostly below freezing point…" This sentence is very vague. Have you measured a profile? Where was the coldest temp measured etc. Please provide more info here.

**Changes: Please read Section 2.2: Experimental procedure. Yes, vertical snow temperature profiles were available to support this statement. Further information were added, see P7L14-17.**

Page 9, Line 17: "To the contrary" Sounds a bit strange. Please revise

**Changes: The sentence was rewritten.**

Page 9, Line 18: Please rethink the title of section 4.2. There can be no rainwater transport WITHIN different flow regimes. Suggestion: "Rainwater transport during the 4 different experiments".

**Changes: Rewritten.**

Page 9, Line 31: What were the snow conditions during those experiments? Please provide some information.

**Changes: Information on LWC and snow temperature have been added.**

Page 10, Line 14: There is now evidence for water refreezing in the snowpack. So please be careful with such statements. Please mention that refreezing of water might have occurred. That the

temperature of the snow pack (mention temp again) is most likely not cold enough that refreezing was a very important process during experiment 1.

**Changes: Our wording is careful: „.... which suggests that some rainwater has been refrozen ...“ does not imply that there is evidence. „Nevertheless, these processes may have been limited to comparably small amount of water“ does further confine our statement. No changes.**

Page 10, Line 15: Please remove the "a" before liquid water.

**Changes: Removed.**

Page 10, Line 19: "in contrast" instead of "to the contrary".

**Changes: Rewritten.**

Page 11, Line 1: The "indeed" here sound a bit strange.

**Changes: The sentence was rewritten. See P12L5.**

Page 11, Lines 9-11. The explanation of the results from the sensitivity analysis should be presented in the results section and explained there in more detail.

**Changes: The results from the sensitivity analysis are now presented in section 3.2. See P8L15-23. Table 6 was also moved to section 3.2. It is newly referred as Table 5.**

Page 11, Line 19+20: Please present those results in the results section. How similar were the results etc.

**Changes: Detail were added in result section.**

Page 11, Line 24: deuterium enriched water

**Changes: Rewritten.**

Page 11, Line 25. There is no evidence form the results that there was refreezing. Please revise this sentence.

**Changes: Rewritten.**

Page 11, Lines 29+35: The word "share" should be replaced by fractions or proportions.

**Changes: Rewritten.**

Page 12, Line 5: ".. is available…"

**Changes: Rewritten.**

Table 1: Dischma en? Please remove.

**Changes: Removed.**

Figure 4: It is hardly possible to distinguish between the colors of the different lines (snow average, rain) here. Again you clearly show here that there was melt prior to the sprinkling in experiment 1. So please mention this in the text!

**Changes: The Figure 4 was replotted and results are commented in Section 4.3.**

Figure 5: please revise the figure caption: "…during all sprinkling experiments.", for example.

**Changes: Rewritten.**

[revised manuscript text omitted]